# UPLC-Q-TOF/MS-Based Metabolomics Approach Reveals Osthole Intervention in Breast Cancer 4T1 Cells

**DOI:** 10.3390/ijms24021168

**Published:** 2023-01-06

**Authors:** Xiuyun Li, Chenglun Zhang, Enhui Wu, Liang Han, Xiangliang Deng, Zhongfeng Shi

**Affiliations:** 1School of Pharmacy, The Center for Drug Research and Development, Guangdong Pharmaceutical University, Guangzhou 510006, China; 2School of Chinese Medicine, Guangdong Pharmaceutical University, Guangzhou 510006, China; 3School of Health, Guangdong Light and Health Engineering R&D Center, Guangdong Pharmaceutical University, Guangzhou 510006, China; 4School of Chinese Medicine, Guangdong Provincial Key Laboratory of Advanced Drug Delivery, Guangdong Provincial Engineering Center of Topical Precise Drug Delivery System, Guangdong Pharmaceutical University, Guangzhou 510006, China

**Keywords:** osthole, UPLC-Q-TOF/MS, breast cancer 4T1 cell, breast cancer

## Abstract

Osthole (OST) is a simple coumarin derivative with pharmacological effects in many types of cancer cells. However, its role and its mechanism of action in breast cancer 4T1 cells remain unclear. In this study, we explored the effects and potential mechanisms of action of OST in 4T1 cells. The MTT, PI, and Annexin V-FITC/PI methods were used to evaluate the effects of OST-treated and untreated 4T1 cells on viability, cell cycle, and apoptosis, respectively. UPLC-Q-TOF/MS combined with multivariate data analysis was used to screen potential biomarkers relevant to the therapeutic mechanisms of OST. Additionally, mTOR, SREBP1, and FASN protein levels were detected using western blotting in OST-treated and untreated 4T1 cells. OST inhibited 4T1 cell proliferation, blocked the cells from remaining in S-phase, and induced apoptosis. In 4T1 cells, OST mainly affected the phospholipid biosynthesis, methyl histidine metabolism, pyrimidine metabolism, and β-oxidation of very long chain fatty acid pathways, suggesting that metabolic changes related to lipid metabolism-mediated signaling systems were the most influential pathways, possibly via inhibition of mTOR/SREBP1/FASN signaling. Our findings reveal biomarkers with potential therapeutic effects in breast cancer and provide insight into the therapeutic and metabolic mechanisms of OST in 4T1 cells.

## 1. Introduction

Globally, breast cancer is the most common and prevalent malignant tumor in women [1,2] and threatens human health and lives. An estimated 2.3 million new cases were diagnosed in women in 2020, accounting for 11.7% of all cancer cases and leading to 685,000 deaths. Breast cancer has surpassed lung cancer as one of the most common causes of cancer-related deaths [3]. Breast cancer is a metastatic cancer that can typically metastasize from its original site to other tissues and organs, such as the bones, liver, lungs, and brain [4,5], and it is often diagnosed at a late stage because symptoms rarely occur in the early stages of disease [6]. Although great progress has been made in the development of diagnostic tools and treatment strategies, current treatment strategies exhibit limitations such as tumor recurrence and metastasis, drug side effects, and drug resistance [7,8,9]. Breast cancer results from the joint action of multiple risk factors in the body, making it extremely difficult to treat [10]. Therefore, novel anticancer drugs are urgently needed. Notably, traditional Chinese medicine has received increased attention because of its unique advantages, such as its effects on multiple pathways and targets and its less toxic side effects [11].

Osthole (OST), also known as methoxyparsleyol, was the first coumarin derivative obtained from medicinal plant *Cnidium monnieri* (L.) Cusson (Figure 1) [12,13]. OST has anti-cancer effects against many types of cancer such as breast cancer, hepatocellular carcinoma, and kidney cancer [14]. The derivative exerts anti-tumor activity through various mechanisms, such as by inhibiting the STAT3 and TGF-β/Smad signaling pathways, inducing PI3K/Akt, and activating the AMPK/Akt/ERK signaling pathway, which includes cell proliferation and the cell cycle, apoptosis, and cell migration [15]. As a process by which tumor cells can meet their energy requirements for survival and proliferation, metabolic reprogramming is considered a hallmark of cancer [16]. A wide range of other metabolic pathways can also sustain cancer cell growth and migration, including amino acid/lipid metabolism, pentose phosphate pathway, and macromolecular biosynthesis [17]. Abnormalities in lipid metabolism have been described in tumor progression [18], particularly fatty acid synthesis (FAS) and fatty acid oxidation (also known as β-oxidation), which play important roles in the malignant transformation of cancer cells [19]. Fatty acid synthase (FASN), a key enzyme in endogenous fatty acid synthesis, is involved in de novo synthesis of long chain fatty acids, which are subsequently esterified to phospholipids and incorporated into cell membranes for cell proliferation [20]. FASN overexpression is related to poor cancer prognosis; therefore, FASN is considered to be a potential drug target for cancer therapy [21]. Sterol regulatory element binding protein 1 (SREBP1), a regulator of FASN transcription, is involved in regulating FASN expression in the nucleus [22]. Additionally, mTOR is major regulator of cellular metabolism via its ability to regulate SREBP1 [23,24]. However, the molecular mechanism of OST in breast cancer 4T1 cells is unclear.

In recent years, ultra high performance liquid chromatography–quadruple–time-of-flight mass spectrometry (UPLC-Q-TOF/MS) based metabolomics techniques have been applied to characterize changes in small molecular weight metabolites (<1.5 kDa) in cells, tissues, and biofluids under specific conditions because of their high sensitivity and resolution and to identify biomarkers useful as therapeutic targets to reveal the metabolic processes and potential mechanisms of various diseases [25]. These results provide a scientific basis for analyzing the pathophysiological pathways in breast cancer [26]. Metabolic composition analysis has suggested that linoleic acid metabolism [27], amino acid, methionine, and purine metabolism [28] are associated with breast cancer progression. Therefore, metabolomic analysis using UPLC-Q-TOF/MS can help reveal the potential antitumor mechanism of OST in 4T1 cells.

In this study, we employed a cellular metabolomic approach using UPLC-Q-TOF/MS to identify promising functional metabolites associated with 4T1 cells and to reveal the metabolic mechanism of OST against 4T1 cells. First, the viability, cycle, and apoptosis of both OST-treated and untreated 4T1 cells were assessed. Next, UPLC-Q-TOF/MS-based methods were used to screen potential metabolic markers associated with OST-treated 4T1 cells. Functional biomarkers were identified, and the therapeutic mechanisms of OST were examined. Finally, mTOR, SREBP1, and FASN protein expression levels were assessed. This study provides insight into the therapeutic mechanisms of OST.

## 2. Results

### 2.1. Effect of OST on Breast Cancer 4T1 Cell Activity

The anticancer effects of various concentrations of OST (0, 25, 50, 75, 100, 125, and 150 μM) on 4T1 cells was investigated in an MTT assay for 24, 48, and 72 h. The results showed that OST inhibited the proliferation of 4T1 cells in a time- and dose-dependent manner compared to the control group (0 μM) (Figure 2). The 50% inhibitory concentrations (IC_50_) of OST treatment for 24, 48, and 72 h were 525.4, 178.4, and 70.7 μM, respectively, for 4T1 cells. Therefore, 50, 75, and 100 μM (as low (L), medium (M), high (H) concentrations) OST were used to treat 4T1 cells for 24 or 48 h in subsequent experiments.

### 2.2. Effect of OST on Cell Cycle of Breast Cancer 4T1 Cells

Based on the MTT assay results, flow cytometry was used to explore the effect of OST on the cell cycle of 4T1 cells. As the dose of OST increased, the percentage of G1-phase cells decreased slightly compared to that in the control group. In addition, the percentage of S-phase cells increased in an OST dose-dependent manner, whereas the percentage of G2-phase cells decreased significantly (Figure 3A,B). This result indicates that OST dramatically induced S-phase cell cycle arrest in 4T1 cells, thereby inhibiting cell growth (Figure 3).

### 2.3. Effect of OST on Apoptosis of Breast Cancer 4T1 Cells

To investigate whether OST induces apoptosis in 4T1 cells, flow cytometry was performed using Annexin V-FITC/PI staining. After treatment with OST for 24 h, the early apoptosis rates of 4T1 cells were 3.57%, 4.47%, and 5.54%, whereas the late apoptosis rates were 29.6%, 31.2%, and 35.7%, respectively (Figure 4A). The percentage of late apoptotic 4T1 cells clearly increased with increasing concentrations of OST compared to that in the control group (0 μM) (Figure 4B). These results demonstrate that OST induced apoptosis of 4T1 cells in dose-dependent manner.

### 2.4. Metabolomics Analysis

#### 2.4.1. Quality Control Sample Detection and System Stability Analysis

Quality control (QC) samples were used to evaluate the repeatability and stability of the system. The total ion flow diagrams of the QC samples were compared by overlapping the spectra (Figure 5). The retention times and peak intensities of the various peaks detected in the QC samples in both electrospray ionization-positive (ESI^+^) and ESI^−^ modes largely overlapped, indicating that the methods were acceptable.

#### 2.4.2. Metabolite Profiling of 4T1 Cells after OST Treatment

After treating 4T1 cells with OST (0 or 75 μM) for 48 h, UPLC-Q-TOF/MS was performed in both ESI^+^ and ESI^−^ modes. Representative total ion chromatograms of endogenous cellular metabolites were obtained, as shown in Figure 6, which revealed significant differences between groups. The extracted data were subjected to substance resolution in the public mass spectrometry database to confirm the identities and chemical structures of the labeled metabolites. In this untargeted metabolomics study, 215 and 214 metabolites were identified from the ESI^+^ and ESI^−^ data, respectively.

### 2.5. Multivariate Statistical Analysis and Screening to Identify Potential Marker Metabolites

Metabolic differences between the control and OST-treated groups were analyzed using multivariate statistics. The UPLC-Q-TOF/MS data were summarized using principal component analysis (PCA) to identify outliers. PCA (Figure 7A,B) and orthogonal projections to latent structures discriminant analysis (OPLS-DA) (Figure 7C,D) scoring plots revealed good separation between the two groups in both ESI^+^ and ESI^−^ modes, suggesting that OST treatment caused metabolic disturbances in 4T1 cells. The data were examined by *t*-test to filter potential marker metabolites with *p* ≤ 0.05 and variable importance in projection (VIP) ≥ 1.0. The resulting data were subjected to *t*-tests, and 32 potential marker metabolites were screened according to *p* ≤ 0.05 and VIP ≥ 1.0 (Table 1). In addition, log_2_fold-change ≥ 1.0 and *p* ≤ 0.05 were used to detect metabolites showing significant differences in metabolism. OST and PC(18:0/18:1(9Z)) were significantly upregulated (fold-change > 1), whereas linoleic acid, glycocholic acid, PC (14:0/0:0), LysoPC(16:0/0:0), 1-O-(cis-9-octadecenyl)-2-O-acetyl-*sn*-glycero-3-phosphocholine, LysoPC(18:1(9Z)/0:0), oleic acid, acetylcarnitine, LysoPE(16:0/0:0), orotic acid, mucic acid, *N*-acetylaspartylglutamate, 3-hydroxyglutaric acid, benzylphosphonic acid, 3-amino-3-(4-hydroxyphenyl) propionic acid, and phosphatidylglyceride18:2–18:2 were significantly downregulated (fold-change < 1). Among these, LysoPC(16:0/0:0), LysoPC(18:1(9Z)/0:0), PC(P-18:0/18:1(9Z)), linoleic acid, PC(14:0/0:0), 1-oleoyl-2-hydroxy-*sn*-glycero-3-PG (sodium salt), oleic acid, LysoPE(18:0/0:0), LysoPE(16:0/0:0), and phosphatidylglyceride 18:2–18:2 are involved in lipid metabolism. To visually identify potential changes in marker metabolites between groups, we clustered 32 marker metabolites for thermal analysis (Figure 8). Each small square represents a potential metabolite, and its color indicates its expression level. Higher expression levels are indicated as a deeper color (red for upregulation, blue for downregulation). Compared to those in the control groups, the relative levels of five marker metabolites were increased and the levels of 27 marker metabolites were decreased after OST treatment.

### 2.6. Metabolic Pathway Analysis

To evaluate the effect of OST on 4T1 cell metabolism, the MetaboAnalyst5.0 database was used for metabolic pathway analysis of the prominent metabolites listed in Table 1. Metabolic pathways with effect values of *p* < 0.2 were selected as potential critical pathways. The regulation of 4T1 cell metabolism by OST involved 24 metabolic pathways (Figure 9A,B). Among them, phospholipid biosynthesis, methyl histidine metabolism, pyrimidine metabolism, and beta-oxidation of very long chain fatty acids were considered as key metabolic pathways through which OST affected 4T1 cells. The 20 metabolic pathways with *p* > 0.2, including α-linolenic and linoleic acid metabolism, purine metabolism, and oxidation of branched chain fatty acids, are summarized in Table 2.

### 2.7. Effect of OST on mTOR/SREBP1/FASN Pathway Proteins in Breast Cancer 4T1 Cells

We examined the expression of mTOR, SREBP1, and FASN in 4T1 cells. The results of western blotting performed to detect protein expression levels are shown in Figure 10. As expected, mTOR, SREBP1, and FASN protein expression was downregulated after OST treatment of 4T1 cells compared to that in the control (0 μM) (Figure 10A). Furthermore, the protein expression of mTOR, SREBP1, and FASN was downregulated in a concentration-dependent manner as the concentrations of OST increased (Figure 10B). OST may induce downregulation of SREBP1 by inhibiting the expression of mTOR, leading to inhibition of FASN expression, thereby inhibiting fatty acid synthesis and esterification to phospholipids (LysoPC, LysoPE or L-acetylcarnitine), and ultimately promoting apoptosis. These results indicate that OST reduced 4T1 cell viability and S-phase cell block, as well as apoptosis, which may be related to inhibition of the mTOR/SREBP1/FASN pathway.

## 3. Discussion

OST is a simple coumarin derivative that exerts anti-tumor activity against many types of tumors through various mechanisms, including inhibition of tumor cell proliferation, induction of tumor cell cycle arrest, and mediation of tumor cell apoptosis [6,14]. OST also inhibits breast cancer cell lines. However, the pathogenesis of breast cancer is more complicated, and the anti-tumor mechanisms of OST treatment in breast cancer are not well understood. In this study, we investigated the anti-tumor effects and mechanisms of OST on breast cancer 4T1 cells and the possible underlying mechanisms. First, we investigated the inhibitory effect of OST on 4T1 cells. The results showed that OST inhibited the proliferation of 4T1 cells in dose- and time-dependent manners. Additionally, OST blocked the 4T1 cells cycle in S-phase and induced apoptosis. Interestingly, phospholipid biosynthesis, methyl histidine metabolism, beta-oxidation of extra long chain fatty acids and pyrimidine metabolism were the most relevant pathways for the antitumor effects of OST on 4T1 cells. Because phospholipid biosynthesis and β-oxidation of very long chain fatty acids are involved in lipid metabolism via esterification and β-oxidation of fatty acids, respectively [20], our data indicate that OST can regulate lipid metabolism.

Lipid metabolism is the most important pathway in cancer, as it generates substances for cancer cell growth and can regulate cellular signaling pathways through downstream products [29]. In addition, the malignant proliferation of cancer cells requires large amounts of energy and macromolecules, inducing carbohydrates, proteins, lipids, and nucleic acids. Particularly, mTOR is a major regulator of cellular metabolism that promotes cellular anabolism to produce various macromolecules such as lipids, proteins, and nucleic acids [23]. OST inhibits FASN expression in HER2-overexpressing breast cancer cells by regulating the Akt/mTOR pathway [30]. Furthermore, FASN promotes breast cancer metastasis by mediating changes in specific fatty acids [21]. These results suggest that mTOR and FASN can be used as drug targets for breast cancer treatment. Furthermore, SREBP1, a transcription factor, is a transcriptional regulator that controls fatty acid and cholesterol biosynthesis and is associated with cell cycle regulation and apoptosis in cancer cells [21,31]. Inhibiting FASN signaling induces cell cycle arrest and reduced proliferation of breast cancer cells [32]. In this study, we explored the effect of OST on breast cancer 4T1 cells in the mTOR/SREBP1/FASN signaling pathway. Our results suggest that OST induces downregulation of SREBP1 by inhibiting the expression of mTOR, thereby leading to inhibition of FASN expression, which reduces fatty acid synthesis and esterification to phospholipids (LysoPC, LysoPE, or L-acetylcarnitine) and ultimately promotes apoptosis.

Malignant cells have greatly increased levels of total fatty acids during proliferation, migration, and invasion [21]. We examined OST metabolites in 4T1 cells to identify potential biomarkers. We screened 32 potential biomarkers, including osthole, linoleic acid, glycocholic acid, PC (14:0/0:0), LysoPC (16:0/0:0), L-histidine, LysoPC (18:1(9Z)/0:0), oleic acid, D-fructose-1-phosphate, L-acetylcarnitine, PC(P-18:0/18:1(9Z)), LysoPE (16:0/0:0), oxypurinol, glutathione, galactonic acid, orotic acid, inosine 5′-monophosphate (IMP), LysoPE (18:0/0:0), UDP-xylose, uridine 5′-diphosphate, phosphatidylglyceride18:2–18:2, and uric acid. Of these, LysoPC (16:0/0:0) and LysoPE (16:0/0:0) are produced via phospholipid biosynthesis. L-acetylcarnitine is a product of beta-oxidation of very long chain fatty acids and oxidation of branched chain fatty acids; IMP and uric acid are products of purine metabolism, and L-histidine is a product of methyl histidine metabolism. Lysophosphatidylcholines (LysoPCs) are a class of compounds with a constant polar head and are distinguished by their fatty acyl groups [33], which can be formed through oxidation of fatty acids [34] or hydrolysis of phosphatidylcholine by phospholipase A2. They can be used as biomarkers for clinical diagnosis, therapy, and pathophysiological studies [35]. Consequently, LysoPC (16:0/0:0), LysoPE (16:0/0:0), and L-acetylcarnitine may be biomarkers for the diagnosis of breast cancer. As a component of cell membranes, phospholipids are produced by esterification, followed by the involvement of FASN in fatty acid synthesis [20], which affects cell proliferation. In this study, we found that phospholipid (LysoPC and LysoPE) levels decreased significantly after OST treatment, suggesting that OST blocked phospholipid synthesis in 4T1 cells by inhibiting mTOR activation and modulating SREBP1 to inhibit FASN activity, which resulted in inhibition of cell proliferation and cell cycle arrest. Consistent with previous studies [32], the levels of unsaturated fatty acids were reduced when we inhibited the mTOR/SREBP1/FASN pathway, resulting in inhibition of 4T1 cell proliferation and arrest of the cell cycle. FASN may inhibit the development and progression of breast cancer by altering the levels of specific fatty acids (LysoPC (16:0/0:0), LysoPE (16:0/0:0), or L-acetylcarnitine). Some studies reported that linoleic acid and oleic acid promote the growth, metastasis, and invasion of breast cancer cells [21]. In contrast, the levels of linoleic and oleic acids decreased after OST treatment in this study, confirming that OST inhibits the growth of 4T1 cells. These results suggest that OST mediates the esterification of intracellular fatty acids into phospholipids for incorporation into cell membranes through the mTOR/SREBP1/FASN signaling pathway to inhibit cell proliferation, cell cycle arrest, and apoptosis induction.

Nucleotides are composed of a nitrogenous base (purine or pyrimidine), a pentose, and one or more phosphate groups and are involved in cell metabolism and physiological regulation [36]. Notably, IMP is an intracellular precursor of adenosine monophosphate and guanosine monophosphate and acts as an extracellular signaling molecule and plays an important role in intracellular purine metabolism [37]. IMP can be catalyzed by a series of enzymatic reactions that result in the production of uric acid [38]. Orotic acid is a precursor of pyrimidine nucleotide biosynthesis [39] and is related to nucleotide metabolism. In the current study, OST blocked 4T1 cells in S-phase (Figure 3), during which cells mainly undergo DNA replication. Orotic acid levels were significantly downregulated, suggesting a reduction in the nucleotide feedstock and inhibition of S-phase DNA replication. In addition, both IMP and uric acid levels were reduced, and catabolism of purine nucleotides may have occurred in 4T1 cells after OST treatment. Thus, the cycle-blocking effect of OST on 4T1 cells may be related to pyrimidine metabolism and purine nucleotide catabolism. These data suggest that OST diminishes nucleotide synthesis and blocks the growth of 4T1 cells to prevent the development and progression of breast cancer.

## 4. Materials and Methods

### 4.1. Materials

Osthole (>98% purity) was purchased from Shanghai Yuanye Biotechnology Co., Ltd. (Shanghai, China). Dulbecco’s Modified Eagle Medium (DMEM) and phosphate-buffered saline (PBS) were purchased from Gibco (Grand Island, NY, USA). Fetal bovine serum was obtained from Biological Industries (Beit Haemek, Israel). Penicillin streptomycin (100×) was purchased from Moocow Biotechology (Guangzhou, China). Trypsin-EDTA (1×, 0.25%) was purchased from Suzhou New Saimei Biotechnology Co., Ltd. (Suzhou, China). 3-(4,5-Dimethylthiazol-2)-2,5-diphenyltetrazolium bromide salt was purchased from Macklin (Shanghai, China). The Cell Cycle Staining Kit and Annexin V-FITC/PI Apoptosis Kit were purchased from Multi Sciences (Hangzhou, China). Acetonitrile was purchased from Thermo Fisher Scientific (Waltham, MA, USA). Methanol was obtained from Honeywell (Charlotte, NC, USA). Ammonium acetate was purchased from Sigma (St. Louis, MO, USA). RIPA lysis buffer, phenylmethylsulfonyl fluoride, and Fickert Ultrasensitive Enhanced Chemiluminescence luminol solution were purchased from Meilunbio Co., Ltd. (Dalian, China). The standard protein bovine serum albumin, BCA kit, sodium dodecyl sulfate polyacrylamide gel electrophoresis (SDS-PAGE) protein loading buffer (5×), and color pre-stained protein molecular weight standard (10–180 kDa) were purchased from Beyotime (Shanghai, China). High molecular weight pre-stained protein markers were purchased from Biomed (Beijing, China). The anti-mTOR antibody was purchased from Servicebio (Wuhan, China). SREBP1 antibody and goat anti-rabbit immunoglobulin horseradish peroxidase were purchased from Affinity (Jiangsu, China). The Fatty Acid Synthase (C20G5) Rabbit mAb was purchased from Cell Signaling Technology (Danvers, MA, USA). Β-Actin antibody was purchased from Shanghai Po Wan Biotechnology Co., Ltd. (Shanghai, China).

The LK16-HF90 CO_2_ Incubator was purchased from Beijing Zhongxihuada Technology Co., Ltd. (Beijing, China). The SW-CF-2FD Ultra-clean table was purchased from Jiangsu Sujing Antai Co., Ltd. (Jiangsu, China). The Fresco17 high speed refrigerated centrifuge and pipettes (range: 1000, 200, 100, 10 μL) were obtained from Thermo Fisher Scientific. A DHP-9272 electronic analytical balance was purchased from Sartorius Scientific Instrument Co., Ltd. (Göttingen, Germany). A SpectraMax Plus384-plus microplate reader was obtained from Molecular Devices (Sunnyvale, CA, USA). A BD FACSCantoll flow cytometer was purchased from BD Biosciences (Franklin Lakes, NJ, USA). The QuickChemi5200+ Chemiluminescence Imaging System was purchased from Monad Biotechnology Co., Ltd. (Suzhou, China). The Q-TOF 5600 Mass Spectrometer was purchased from AB Sciex Corporation (Framingham, MA, USA). The Nexera Ultra-Performance Liquid Chromatograph was purchased from Shimadzu Corporation (Kyoto, Japan).

### 4.2. Cell Culture

Mouse breast cancer 4T1 cells were provided by the Guangdong Provincial Key Laboratory of New Drug Models. 4T1 cells were cultured in DMEM containing 10% fetal bovine serum, 100 U/mL penicillin, and 100 μg/mL streptomycin, in a constant temperature cell culture incubator at 37 °C with 5% CO_2_. The cells were digested and passaged with 0.25% trypsin (without EDTA) after 2–3 days of adherent culture. Cells in the exponential growth phase were used in all experiments.

### 4.3. Cell Viability Assay

We performed an MTT assay to examine the viability of 4T1 cells treated with OST. Briefly, 4T1 cells (4000 cells/well) were seeded into 96-well plates to culture for 24 h, followed by treatment of the cells with different concentrations of OST (0, 25, 50, 75, 100, 125, and 150 μM). After treatment for 24, 48, and 72 h, 100 μL DMEM basal culture and 15 μL MTT solution (5 mg/mL in PBS) were added to each well and incubated for 4 h at 37 °C. Finally, the culture supernatant in the wells was aspirated, and 150 μL of dimethyl sulfoxide was added to each well and shaken for 10 min at room temperature to melt the crystals. Absorbance was measured at 490 nm using a multifunctional enzyme marker. The percentage of cell viability was calculated from the mean optical density (OD) values of the wells using the following formula: cell viability (%) = (OD _Test_ /OD _Control_) × 100.

### 4.4. Cell Cycle Assay

To analyze the effect of OST on the 4T1 cell cycle, the DNA content of 4T1 cells was measured using flow cytometry. 4T1 cells (3 × 10^5^ cells/mL, 2 mL/well) were seeded into 6-well plates and incubated for 24 h. The cells were collected after treatment with complete medium or different concentrations of OST (0, 50, 75, and 100 μM) for 24 h. The cells were washed with pre-cold PBS and incubated with 1mL of DNA staining solution and 10 μL of permeabilization solution for 30 min at room temperature in the dark. The cell cycle was analyzed using flow cytometry. The data were analyzed using ModFit LT software.

### 4.5. Cell Apoptosis Assay

Apoptosis was detected using flow cytometry. 4T1 cells (3 × 10^5^ cells/mL, 2 mL/well) were added to 6-well plates and cultured for 24 h. After treatment with different concentrations of OST (0, 50, 75, and 100 μM) for 24 h, the cells were collected, washed with pre-cold PBS, and resuspended in 500 μL of 1× binding buffer. Next, 5 μL of Annexin V-FITC and 10 μL of PI were added to the cell buffer and incubated for 5 min in the dark. Cell apoptosis was analyzed using flow cytometry. The data were analyzed using FlowJo software (TreeStar, Ashland, OR, USA).

### 4.6. Sample Preparation for Metabolomics

4T1 cells were cultured in 6-well plates (5 × 10^5^/mL, 2 mL/well) for 24 h and treated with OST (0 or 75 μM) for 48 h. The OST treated group (75 μM) and control group (0 μM) were subjected to three repeated experiments. 4T1 cells were washed with pre-cold PBS and saline (0.9% NaCl), scraped with 1 mL of pre-cold ultrapure water, transferred to 2.0 mL tubes, and stored at −80 °C until use (approximately 10^7^ cells per sample). For the experiment, the samples were defrosted on ice; 100 μL of this sample was added to a new tube, to which 1000 μL of extraction solution (methanol: acetonitrile: water = 2:2:1, *v*/*v*) was mixed and vortexed for 15 s. The samples were ultrasonicated in an ice-water bath for 10 min and snap-frozen in liquid nitrogen for 1 min; these steps were repeated three times. The samples were incubated at −20 °C for 1 h, followed by centrifugation at 13,000 rpm for 15 min at 4 °C. The supernatant was extracted, transferred to new tubes, and dried under a nitrogen stream. The dried samples were dissolved and shaken vigorously for 30 s in 100 μL acetonitrile: water = 1:1 (*v*/*v*) solution before sonication in an ice-cold water bath for 10 min and centrifugation at 13,000 rpm for 15 min at 4 °C. The supernatants were carefully removed and prepared for the assay. QC samples were prepared in parallel by mixing 20 μL of each sample supernatant to verify the stability of the LC-MS/MS system. The QC samples were injected three consecutive times prior to the study.

### 4.7. UPLC-Q-TOF/MS Conditions

Metabolomic analysis was performed using an UPLC system coupled with a mass spectrometer. A UPLC BEH Amide column (1.7 μm, 2.1 × 100 mm, Waters, Milford, MA, USA) was used for separation. The column was maintained at 55 °C at a flow rate of 0.3 mL·min^−1^. The gradient eluents consisted of 100% H_2_O, 25 mM CH_3_COONH_4_ + 25 mM NH_4_OH (A) and 100% acetonitrile (B). The ion-mode gradient elution conditions were as follows: 85% B at 0–1 min; 65% B at 1–12 min; 40% B at 12–12.1 min; 40% B at 12.1–15 min; 85% B at 15–15.1 min; 85% B at 15.1–20 min. The sample injection volume was 5 μL. The automatic samplers were maintained at 4 °C.

An ESI source was used for mass spectrometry analysis. The ESI source operation parameters were as follows: source temperature, 600 °C; ion source voltage, 5500 V or −4500 V; air curtain gas, 20 psi; atomization gas and auxiliary gas, 60 psi. Scanning was performed by multiple reaction monitoring.

### 4.8. Metabolomics Data Analysis

Spectral data were collected using UPLC-Q-TOF/MS technology and compared in both XCMS and VGDB for peak alignment, peak retention time, peak area selection, and quality (mass-to-charge ratio). The resultant data were imported into SIMCA14.1, and the clustering information and significant variables were captured using PCA and OPLS-DA. The OPLS-DA model was used to calculate the VIP values for each variable in the cell samples, and differential metabolites between the groups at various time points were screened based on VIP ≥ 1.0. Data were normalized and log-transformed, and *p*-values were calculated using *t*-tests. Metabolites with VIP ≥ 1 and *p* ≤ 0.05 were selected as final differential metabolites, which were validated using the Human Metabolome Databases (http://www.hmdb.ca/) (accessed on 15 June 2022), METLIN (https://metlin.scripps.edu/) (accessed on 15 June 2022), and Massbank (https://massbank.eu/) (accessed on 15 June 2022), as well as other database models. The validated differential metabolites were entered into the MetaboAnalyst5.0 database (https://www.metaboanalyst.ca/) (accessed on 16 July 2022) for metabolic pathway analysis.

### 4.9. Western Blot Analysis

After treatment with various OST concentrations (0, 50, 75, and 100 μM) for 48 h, the 4T1 cells were washed with cold PBS. The cells were lysed with RIPA lysis buffer on ice for 30 min and then scraped. The cell lysates were centrifuged at 13,000 rpm at 4 °C for 20 min to collect the supernatant. The protein concentration in the supernatant was determined using a BCA protein assay kit. Proteins were heated at 95 °C for 5 min in SDS-PAGE protein loading buffer for denaturation. Equal amounts of protein were separated by SDS-PAGE at 6–12%, which were transferred to PVDF membranes and blocked with Tris-buffered saline containing Tween 20 and 5% skimmed milk for 1 h. The blots were washed with 1× Tris-buffered saline containing Tween 20 and incubated with mTOR (1:1000), SREBP1 (1:1000), FASN (1:1000), and β-actin at 4 °C overnight. The next day, the blots were washed and exposed to goat anti-rabbit immunoglobulin horseradish for 2 h at room temperature. The blots were washed and activated with enhanced chemiluminescence solution. The data were quantified using a chemiluminescent imaging system. The gray scale values of the blots were analyzed using ImageJ software (NIH, Bethesda, MD, USA).

### 4.10. Statistical Analysis

Data analysis was performed using GraphPad Prism 9.0 (GraphPad, Inc., La Jolla, CA, USA). All data are expressed as the mean ± standard deviation and were obtained from at least three independent experiments. Differences between experimental groups were analyzed using Student’s *t*-test. Statistical significance was set at *p* < 0.05.

## 5. Conclusions

OST exhibits antitumor effects in breast cancer 4T1 cells. We showed that OST inhibits fatty acid and nucleotide levels, thereby inhibiting 4T1 cell proliferation, blocking the cell cycle and inducing apoptosis. The inhibitory effects of OST on 4T1 cells mainly involved phospholipid biosynthesis, methylhistidine metabolism, pyrimidine metabolism, and beta-oxidation of ultra long chain fatty acids. Of these, OST may modulate the activities of mTOR, SREBP1, and FASN. Because lipid metabolism is a key feature of cancer-specific metabolism, enzymes associated with this process were further evaluated. These results suggest that OST induces downregulation of SREBP1 through inhibition of mTOR expression, leading to inhibition of FASN expression to interfere with lipid metabolic pathways such as phospholipid biosynthesis, beta oxidation of very long chain fatty acids, and oxidation of branched chain fatty acids in 4T1 cells, inhibiting the de novo synthesis and esterification of long chain fatty acids into phospholipids (LysoPC, LysoPE, or L-acetylcarnitine) into the cell membrane, thereby inhibiting 4T1 cell proliferation, blocking the cell cycle and promoting apoptosis. OST shows potential as a drug for treating breast cancer, acting through a variety of metabolic pathways that regulate amino acid, fatty acid, and nucleotide synthesis. Our findings reveal potential biomarkers that can contribute to a more comprehensive understanding of the anti-tumor mechanisms of OST in 4T1 cells.

## Figures and Tables

**Figure 1 ijms-24-01168-f001:**
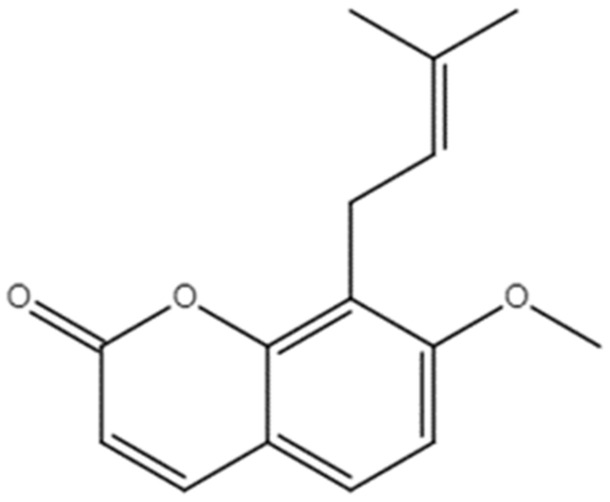
Chemical structure of Osthole [7-Methoxy-8-(3-methyl-2-butenyl) coumarin].

**Figure 2 ijms-24-01168-f002:**
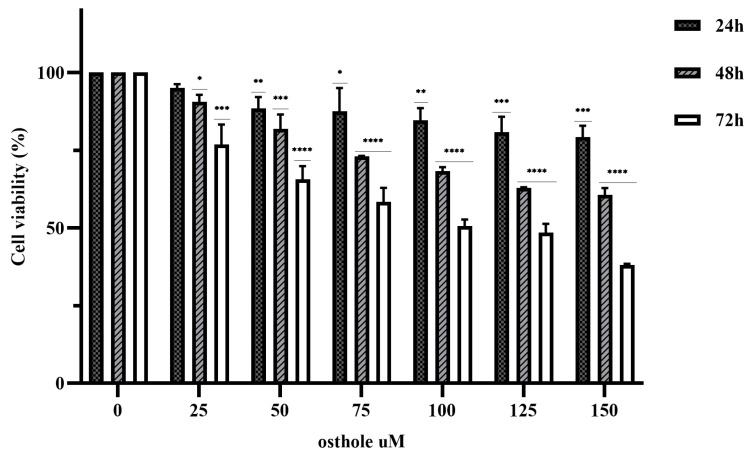
Osthole (OST) inhibits the proliferation of 4T1 cells. 4T1 cells were incubated with OST (0, 25, 50,75, 100, 125, and 150 μM) for 24, 48, and 72 h. Cell viability was determined in an MTT assay. Compared to that in the control group, Mean ± SD, N = 3, * *p* < 0.05, ** *p* < 0.01, *** *p* < 0.001, **** *p* < 0.0001.

**Figure 3 ijms-24-01168-f003:**
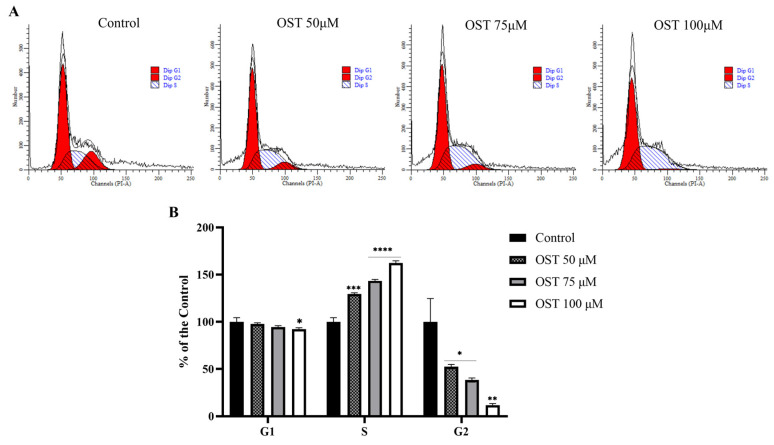
Effect of osthole (OST) on 4T1 cell cycle distribution. (**A**) 4T1 cells were treated with (0, 50, 75, and 100 μM) OST for 24 h. DNA content was analyzed using flow cytometry with propidium iodide (PI) staining. (**B**) DNA content in each phase of the cell cycle (G1, S, G2) after OST treatment. Compared to that in the control group, Mean ± SD, N = 3, * *p* < 0.05, ** *p* < 0.01, *** *p* < 0.001, **** *p* < 0.0001.

**Figure 4 ijms-24-01168-f004:**
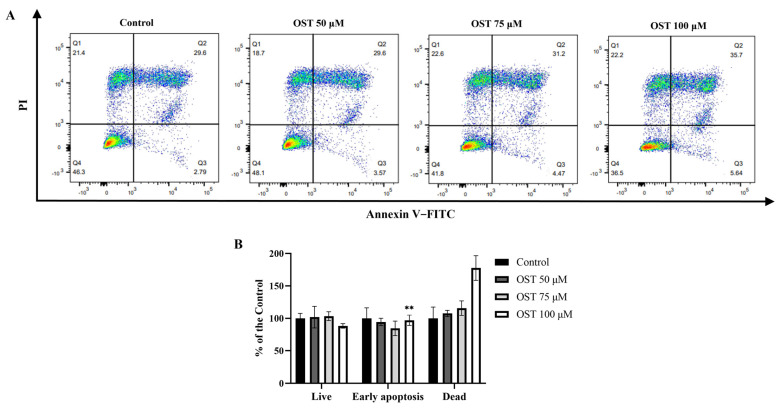
Effect of osthole (OST) on apoptosis of 4T1 cells. (**A**) 4T1 cells were treated with OST (0, 50, 75, and 100 μM) for 24 h. Apoptosis and cell death induction were detected in an Annexin VFITC /PI double staining assay. (**B**) Mean percentage of cells in each quadrant of 4T1 cells after OST treatment. Viable cells were considered as FITC Annexin V^−^/PI^−^; early apoptosis cells were considered as FITC Annexin V^+^/PI^−^; and late apoptosis/dead cells were identified as FITC Annexin V^+^/PI^+^. Compared to that in the control group, Mean ± SD, N = 3, ** *p* < 0.01.

**Figure 5 ijms-24-01168-f005:**
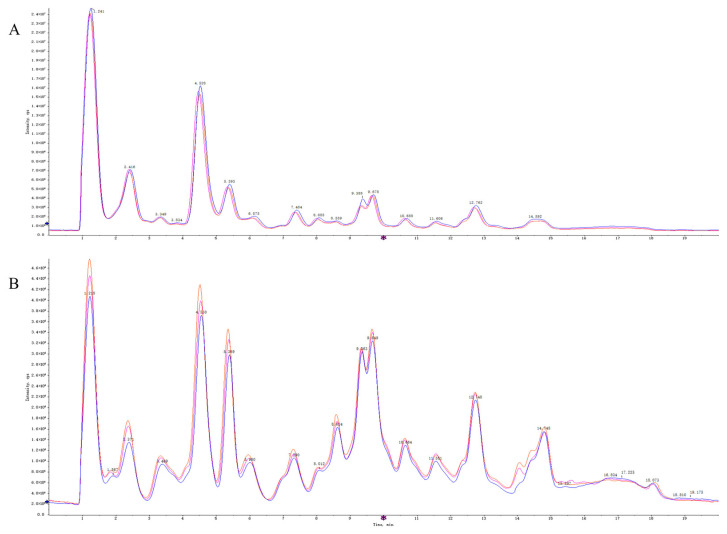
The TIC overlaps of QC samples of in ESI^+^ mode (**A**) and ESI^−^ mode (**B**).

**Figure 6 ijms-24-01168-f006:**
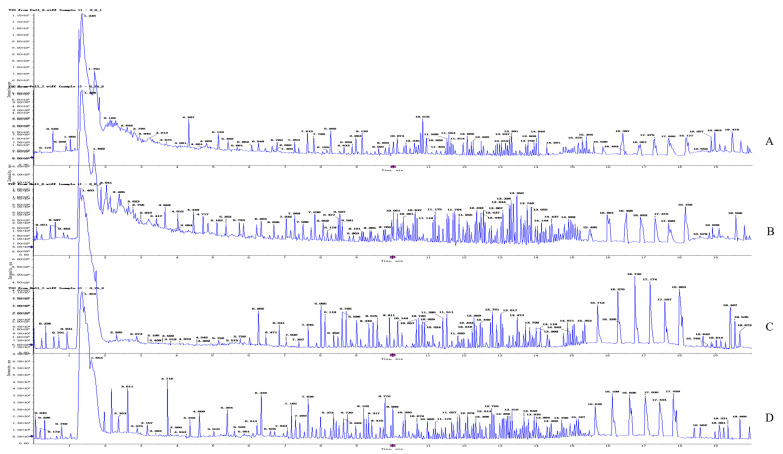
Representative total ion chromatogram obtained for metabolite profiles in 4T1 cells after osthole (OST) treatment. ESI^+^ mode: (**A**) control group; (**B**) 75 μM OST; ESI^−^ mode: (**C**) control group; (**D**) 75 μM OST.

**Figure 7 ijms-24-01168-f007:**
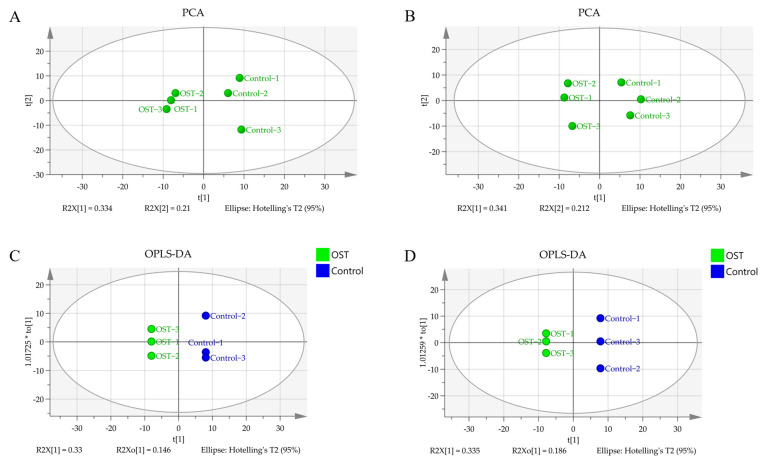
Principal component analysis scoring plots for endogenous metabolites of 4T1 cells in electrospray ionization-positive (ESI^+^) mode (**A**) and ESI^−^ mode (**B**), orthogonal projections to latent structures discriminant analysis (OPLS-DA) scoring plots for ESI^+^ mode (**C**) and ESI^−^ mode (**D**).

**Figure 8 ijms-24-01168-f008:**
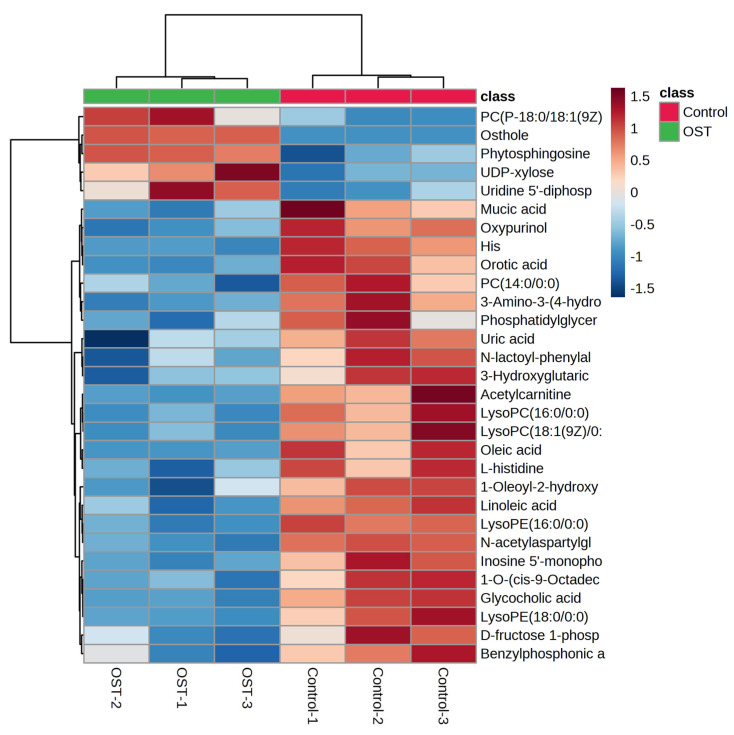
Changes in relative content of potential biomarker 4T1 cells after osthole (OST) treatment.

**Figure 9 ijms-24-01168-f009:**
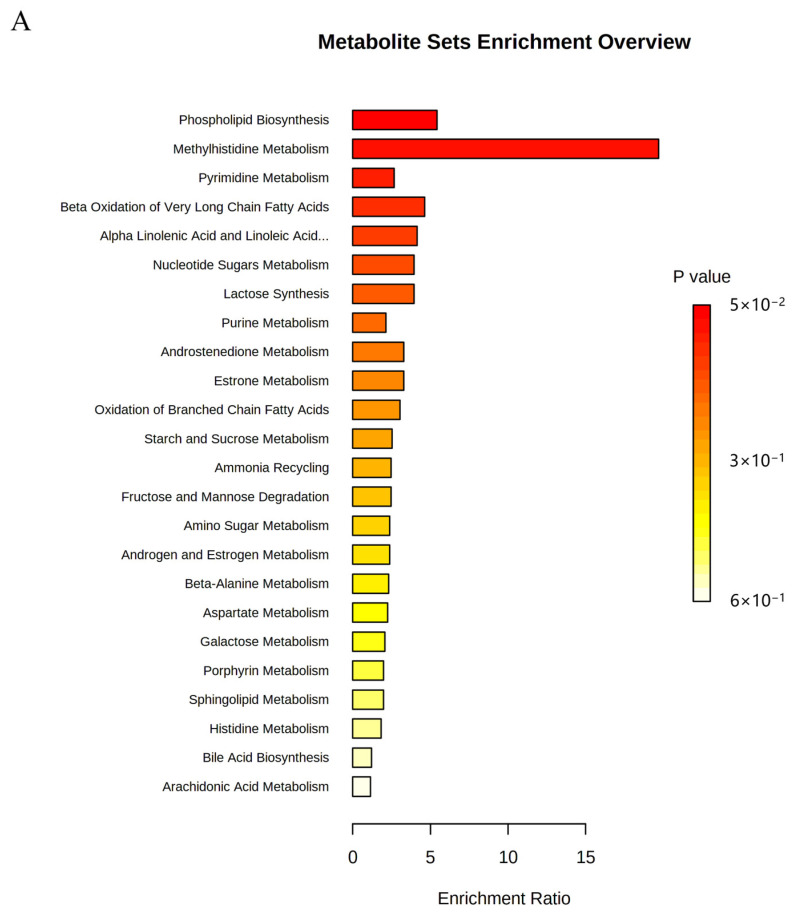
Metabolic pathway enrichment analysis of differential metabolites conducted based on the Kyoto Encyclopedia of Genes and Genomes (KEGG) database. (**A**) Bar chart and (**B**) Bubble chart.

**Figure 10 ijms-24-01168-f010:**
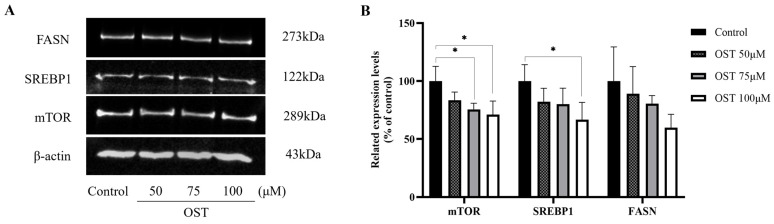
Effect of osthole (OST) on the expression of mTOR, SREBP1, and FASN in 4T1 cells. (**A**) 4T1 cells were treated with OST (0, 50, 75, and 100 μM) for 48 h. The expression levels of mTOR, SREBP1, and FASN in the cells were detected using western blotting. (**B**) Quantitative analysis using ImageJ software of western blot signal intensity expression relative to β-actin loading. Data show the mean ± SD (three independent experiments) compared to that in the control group (0 μM). * *p* < 0.05.

**Table 1 ijms-24-01168-t001:** Altered metabolites of 4T1 cells after osthole (OST) treatment in both electrospray ionization-positive (ESI^+^) and ESI^−^ mode.

NO	M/Z	Compound Name	Chemical Formula	*p*-Value	Trend
1	245.1170	Osthole	C_15_H_16_O_3_	0.0001	↑
2	521.3430	Linoleic acid	C_18_H_32_O_2_	0.0048	↓
3	466.3285	Glycocholic acid	C_26_H_43_NO_6_	0.0061	↓
4	468.3065	PC (14:0/0:0)	C_22_H_46_NO_7_P	0.0151	↓
5	496.3388	LysoPC (16:0/0:0)	C_24_H_50_NO_7_P	0.0151	↓
6	318.2995	Phytosphingosine	C_18_H_39_NO_3_	0.0195	─
7	550.3854	1-O-(cis-9-octadecenyl)-2-O-acetyl-*sn*-glycero-3-phosphocholine	C_28_H_57_NO_7_P^+^	0.0203	↓
8	522.3544	LysoPC (18:1(9Z)/0:0)	C_26_H_52_NO_7_P	0.0246	↓
9	521.3430	Oleic acid	C_18_H_34_O_2_	0.0254	↓
10	283.0186	D-fructose-1-phosphate	C_6_H_13_O_9_P	0.0361	─
11	204.1222	L-acetylcarnitine	C_9_H_18_NO_4_	0.0407	↓
12	772.6059	PC (P-18:0/18:1(9Z))	C_44_H_86_NO_7_P	0.0445	↑
13	452.2755	LysoPE (16:0/0:0)	C_21_H_44_NO_7_P	0.0005	↓
14	303.2319	N-acetylaspartylglutamate (NAAG)	C_11_H_16_N_2_O_8_	0.0011	↓
15	151.0257	Oxypurinol	C_5_H_4_N_4_O_2_	0.0017	─
16	154.0617	His	C_6_H_9_N_3_O_2_	0.0049	─
17	306.0765	Glutathione	C_10_H_17_N_3_O_6_S	0.0071	─
18	195.0508	Galactonic acid	C_6_H_12_O_7_	0.0083	─
19	180.0660	3-amino-3-(4-hydroxyphenyl) propionic acid	C_9_H_11_NO_3_	0.0092	↓
20	154.0617	L-histidine	C_6_H_9_N_3_O_2_	0.0098	─
21	155.0097	Orotic acid	C_5_H_4_N_2_O_4_	0.0156	↓
22	347.1826	Inosine 5′-monophosphate (IMP)	C_10_H_13_N_4_O_8_P	0.0173	─
23	236.0922	N-lactoyl-phenylalanine	C_12_H_15_NO_4_	0.0210	─
24	147.0295	3-hydroxyglutaric acid	C_5_H_8_O_5_	0.0236	↓
25	509.2864	1-oleoyl-2-hydroxy-*sn*-glycero-3-PG (sodium salt)	C_24_H_47_NaO_9_P	0.0254	─
26	480.3070	LysoPE (18:0/0:0)	C_23_H_48_NO_7_P	0.0298	─
27	535.0338	UDP-xylose	C_14_H_22_N_2_O_16_P_2_	0.0302	─
28	171.0260	Benzylphosphonic acid	C_7_H_9_O_3_P	0.0315	↓
29	209.0300	Mucic acid	C_6_H_10_O_8_	0.0372	↓
30	402.9918	Uridine 5′-diphosphate	C_9_H_14_N_2_O_12_P_2_	0.0390	─
31	771.5151	Phosphatidylglyceride18:2–18:2	C_42_H_75_O_10_P	0.0459	↓
32	167.0204	Uric acid	C_5_H_4_N_4_O_3_	0.0491	─

**Table 2 ijms-24-01168-t002:** Statistical analysis results of the main metabolites in 4T1 cells.

NO	Pathway Name	Total	Expected	Hits	Raw *p*
1	Phospholipid Biosynthesis	29	0.368	2	0.0498
2	Methyl Histidine Metabolism	4	0.051	1	0.0499
3	Pyrimidine Metabolism	59	0.749	2	0.169
4	Beta Oxidation of Very Long Chain Fatty Acids	17	0.216	1	0.197
5	Alpha Linolenic Acid and Linoleic Acid Metabolism	19	0.241	1	0.217
6	Nucleotide Sugar Metabolism	20	0.254	1	0.227
7	Lactose Synthesis	20	0.254	1	0.227
8	Purine Metabolism	74	0.939	2	0.241
9	Androstenedione Metabolism	24	0.305	1	0.267
10	Estrone Metabolism	24	0.305	1	0.267
11	Oxidation of Branched Chain Fatty Acids	26	0.330	1	0.286
12	Starch and Sucrose Metabolism	31	0.394	1	0.331
13	Ammonia Recycling	32	0.406	1	0.340
14	Fructose and Mannose Degradation	32	0.406	1	0.340
15	Amino Sugar Metabolism	33	0.419	1	0.348
16	Androgen and Estrogen Metabolism	33	0.419	1	0.348
17	Beta-Alanine Metabolism	34	0.432	1	0.357
18	Aspartate Metabolism	35	0.444	1	0.365
19	Galactose Metabolism	38	0.482	1	0.390
20	Porphyrin Metabolism	40	0.508	1	0.406
21	Sphingolipid Metabolism	40	0.508	1	0.406
22	Histidine Metabolism	43	0.546	1	0.429
23	Bile Acid Biosynthesis	65	0.825	1	0.576
24	Arachidonic Acid Metabolism	69	0.876	1	0.598

## Data Availability

The data presented in this study are available within the article.

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
