# Peer review of "UPLC-Q-TOF/MS-Based Metabolomics Approach Reveals Osthole Intervention in Breast Cancer 4T1 Cells"

_ijms, 2023, doi:10.3390/ijms24021168_

Round 1

Reviewer 1 Report

I thank the authors of the manuscript entitled" A UPLC-Q-TOF/MS-based metabolomics approach reveals Osthole intervention in breast cancer 4T1 cells" for their efforts, though several issues are there.

Language really needs substantial improvements as the manuscript in its present form is not readable and understandable.

You measured the cell cycle analysis and apoptosis in different concentrations of the drug but you did not mention the time point and justify why you chose a time point and leave the others.

you mentioned in the title that OST has interfered with breast cancer cells but you did not mention how this happens and the mechanism of action by which the drug did that.

I thank you again for your work, but actually, this manuscript needs further revision before being suitable for publication. 

Author Response

请参阅附件。

Reviewer 2 Report

Review for the manuscript ijms-2039987.

      The manuscript ijms-2039987 by Li et al reports interesting data about antiproliferation effects of osthole, coumarin-like compound from the plant Cnidium monnieri (L.), the genus Cnidium, against breast cancer cells 4T1. The results of this important work is based on modern UPLC-Q-TOF/MS metabolomics techniques and showed that OST inhibited the proliferation of 4T1 cells, blocked the cells from staying in S-phase as well as induced apoptosis. These findings may provide biomarkers of potential therapeutic effects in breast cancer and foster insight into the therapeutic and metabolic mechanisms of OST on 4T1 cells.

The manuscript ijms-2039987 is perfectly structured, well written and needs only few minor corrections:

line 47, please correct “from the genus osthole”to “from the genus Cnidium”; (osthole is a compound, not the genus of plants);

line 62, please correct font “de novo” to “de novo” (italica);

line 230, please correct “cancer lines. however” to “cancer lines. However”

lines 270, 271, 272, 273, 282, 283, 293, please correct biomarker names “D-Fructose” to “D-fructose”, “L-Acetylcarnitine” to “L-acetylcarnitine”, and other similar typos.

References

Please correct references 1, 9, 23, 35 and 38. Use comma after the author surname (“Siegel R.L.”to

“Siegel, R.L.” and similar corections for other mentioned references.

After all listed minor corrections will be done, the current manuscript

can be accepted for  publication.
